# Multiblock Copolymers of Norbornene and Cyclododecene: Chain Structure and Properties

**DOI:** 10.3390/polym13111756

**Published:** 2021-05-27

**Authors:** Yulia I. Denisova, Georgiy A. Shandryuk, Marianna P. Arinina, Ivan S. Levin, Vsevolod A. Zhigarev, Maria L. Gringolts, Eugene Sh. Finkelshtein, Alexander Ya. Malkin, Yaroslav V. Kudryavtsev

**Affiliations:** 1Topchiev Institute of Petrochemical Synthesis, Russian Academy of Sciences, Leninskii pr. 29, 119991 Moscow, Russia; denisova@ips.ac.ru (Y.I.D.); gosha@ips.ac.ru (G.A.S.); arinina.marianna@ips.ac.ru (M.P.A.); levin@ips.ac.ru (I.S.L.); zhigarev@ips.ac.ru (V.A.Z.); gringol@ips.ac.ru (M.L.G.); fin314@gmail.com (E.S.F.); alex_malkin@mig.phys.msu.ru (A.Y.M.); 2Frumkin Institute of Physical Chemistry and Electrochemistry, Russian Academy of Sciences, Leninskii pr. 31, 119071 Moscow, Russia

**Keywords:** macromolecular cross-metathesis, multiblock copolymer, crystallinity, thermal fractionation

## Abstract

We investigate the structure–property relations of the multiblock copolymers of norbornene with cyclododecene synthesized via the macromolecular cross-metathesis reaction between amorphous polynorbornene and semicrystalline polydodecenamer in the presence of the first-generation Grubbs catalyst. By adjusting the reaction time, catalyst amount, and composition of the initial system, we obtain a set of statistical multiblock copolymers that differ in the composition and average length of norbornene and dodecenylene unit sequences. Structural, thermal, and mechanical characterization of the copolymers with NMR, XRD, DSC (including thermal fractionation by successive self-nucleation and annealing), and rotational rheology allows us to relate the reaction conditions to the average length of crystallizable unit sequences, thicknesses of corresponding lamellas, and temperatures of their melting. We demonstrate that isolated dodecenylene units can be incorporated into crystalline lamellas so that even nearly random copolymers should retain crystallinity. Weak high-temperature endotherms observed in the multiblock copolymers of norbornene with cyclododecene and other cycloolefins could indicate that the corresponding systems are microphase-separated in the melt state.

## 1. Introduction

Chain structure in copolymers is usually characterized by the average lengths of blocks consisting of chemically or structurally different monomer units. Even a moderate tendency toward blockiness in the monomer unit distribution can result in considerably different properties relative to completely random polymers of the same composition [1,2,3,4,5]. Recent reviews [6,7,8,9,10] discuss a number of well-established techniques nowadays available for the synthesis of multiblock copolymers. Depending on the synthetic conditions, the monomer unit order in copolymer chains varies from fully determined in the so-called sequence-defined polymers [11] to statistical in the products of macromolecular reactions, such as post-modification, end coupling, or interchain exchange [12].

Growing interest in multiblock copolymers is driven by the expansion of their practical application scope, which by now includes templates, packaging and shape memory materials, ion-exchange membranes, energy storage and photonic media, adhesives, coatings, impact modifiers, emulsifiers, Janus nanoparticles, and gene and drug carriers [13,14,15,16,17]. All the rich variety of properties is based on the ability of block copolymers to self-assemble into different morphologies, in which monomer unit interactions are balanced by their connectivity in polymer chains [18,19]. Being inferior to diblock copolymers in the accuracy and alignment of domain boundaries, multiblocks demonstrate better mechanical properties, biocompatibility, biodegradability, compatibilizing ability, and tendency to form bicontinuous phases for effective ionic and molecular transport [20,21,22,23,24,25].

Macromolecular cross-metathesis (MCM) is an interchange-type reaction between polymers containing C=C bonds in the backbone [9]. When two homopolymers exchange their segments, it leads to the formation of their copolymer with a blocky structure, the blocks being gradually shortened with the extent of the reaction. MCM attracts special interest in situations where a direct copolymerization of two monomers is for some reason complicated or even impossible [26,27,28,29,30,31]. Our interest was initially focused on using MCM for the synthesis of multiblock copolymers of norbornene and cyclooctene [32]. This reaction starts from commercially available polynorbornene and polyoctenamer, thus circumventing the difficulties of copolymerizing monomers with considerably different polymerization activities. As interchange reactions do not affect the type of monomer units, the copolymer properties are determined by the degree of blockiness, which can be effectively regulated using the kinetic model of MCM [33] to adjust the reaction time, catalyst amount, and initial blend composition. A number of functional groups can be introduced either by inserting them into the monomers used to obtain starting homopolymers or by post-modification of the resulting multiblock copolymers [34,35,36,37].

In a recent paper [38], we demonstrated the possibility to use MCM for the synthesis of statistical multiblock copolymers of norbornene (NB) and cyclododecene (CDD). CDD is one of the widely available petrochemical cycloolefins. The product of its polymerization, polydodecenamer (PCDD), is characterized by a higher degree of crystallinity and melting point compared with other polyalkenamers [39,40,41,42]. It is used as an additive to elastomers, primarily to increase the hardness of styrene–butadiene rubber [43] and other vulcanizates. The activity of CDD in metathesis polymerization is even lower than that of cyclooctene, and the cometathesis of CDD and NB is difficult due to the significant difference in the cycle strain energy (27.2 kcal/mol for NB and 11.1 kcal/mol for CDD) [44]. At the same time, CDD copolymers and products of their further modification involving double bonds can be used in materials with shape memory that possess melting and glass transition temperature transitions in various ranges [45], sparsely but regularly substituted polyethylene [46], multipurpose composites, and materials with enhanced gas permeability [47].

In this study, we investigate the structure–property relationships of NB–CDD copolymers of different composition and the degree of blockiness. Application of high-resolution nuclear magnetic resonance (NMR) makes it possible to calculate the composition, *cis/trans* ratio, and average block lengths of the copolymers obtained via the MCM reaction under different conditions. Combination of X-ray diffraction (XRD) and differential scanning calorimetry (DSC) allows us to characterize the copolymer crystallinity, in particular, to estimate the average, maximum, and minimum thicknesses of crystalline lamellas. DSC and oscillatory rheological experiments shed some light on the copolymer morphology above the full amorphization temperature. Application of the DSC-based fractionation technique known as the successive self-nucleation and annealing (SSA) method [48] deserves special attention due to its growing popularity [49], including application to multiblock copolymers [50,51]. When a sample undergoes several heating–annealing–cooling cycles, a stepwise decrease in the annealing temperature leads to the formation of a set of crystalline lamellas with different average thicknesses. Although this process does not allow one to physically isolate them, the final melting thermogram contains information on their temperature and enthalpy of melting, which enables the calculation of the lamellar thickness distribution in the crystalline part of the sample. The implementation of SSA to the NB–COE copolymers [51] helped us to demonstrate that a subtle difference in the average block length can lead to a pronounced shift in the thermodynamic, rheological, and relaxation behavior in solution and bulk.

The paper is organized as follows. We start with describing the synthesis and characterization of PCDD and PNB homopolymers, PCDD/PNB blend, and seven NB–CDD copolymers of different composition and degree of blockiness, including a detailed investigation of the chain microstructure by NMR. Then we continue with presenting the results of DSC measurements on all mentioned systems with and without SSA thermal fractionation. Finally, we discuss the correlations between chain structure and crystallinity, compare NB–CDD and previously studied NB–COE copolymers, analyze an interesting but subtle endothermic effect in the melt state, and end the paper with a summary of our findings.

## 2. Materials and Methods

Polynorbornene (poly(1,3-cyclopentylenevinylene), PNB) and polydodecenamer (poly(1-dodecenylene), PCDD) were synthesized by ring-opening metathesis polymerization (ROMP) in the presence of the first (G1)- and second (G2)-generation Grubbs catalysts, respectively (Scheme 1a, see [38] for details). In order to remove the catalyst residues, 0.23 M solutions of PCDD and PNB in chloroform were passed through a column with SiO_2_ (SiO_2_/polymer = 4:1, wt/wt) and precipitated in ethanol, decanted, washed with several portions of ethanol, and dried under reduced pressure at room temperature until constant mass. The reaction yield was 97% for PNB and 96% for PCDD.

Because of the limited solubility of PCDD in organic solvents at room temperature, the cross-metathesis of PNB with PCDD (Scheme 1b) was carried out in CHCl_3_ under inert atmosphere at 40 °C. Upon dissolving the homopolymers, the solution of the G1 catalyst in CHCl_3_ was added. The mixture was continuously stirred for the required time. Cross-metathesis was terminated by the addition of ethyl vinyl ether to the reaction mixture (ethyl vinyl ether/catalyst = 500:1 mol/mol) and stirring for 30–40 min at 40 °C. Polymer solutions were stabilized by adding 0.01% (relative to the polymer mass) 2,2′-methylene-bis(6-tert-butyl-4-methylphenol) (Sigma-Aldrich, St. Louis, MO, USA) as an oxidation inhibitor and stirring for the next 30–40 min, followed by concentrating in vacuum (Table 1, sample C4) or precipitation in ethanol by adding 0.5 g/L of the above inhibitor and drying as described above (Table 1, samples C1–C3 and C5–C7). In that manner, seven NB–CDD copolymers corresponding to the different reaction time and composition of the initial homopolymer mixture were obtained. An equimolar blend PCDD/PNB was prepared by dissolving the homopolymers and 0.01% (relative to the polymer mass) of the inhibitor in CHCl_3_ and precipitating the blend in ethanol as described above.

The molar mass of the polymers was determined by gel permeation chromatography (GPC) on a Waters (Milford, USA) high-pressure chromatograph equipped with a refractometric detector and Microgel mix 1–5 μm 300 × 7.8 mm^2^ Waters Styragel HR 5E column, with tetrahydrofuran as a solvent, a flow rate of 1 mL/min, a sample volume of 100 μL, a column temperature of 313 K, and a sample concentration of 1 mg/mL. The average molar mass and its dispersity (*Ð*) were calculated by a standard procedure relative to polystyrene standards.

Nuclear magnetic resonance (NMR) measurements were carried out at room temperature using a Bruker (Billerica, MA, USA) Avance™ 600 NMR spectrometer operating at 150.93 MHz (^13^C NMR); CDCl_3_ (Aldrich, Milwaukee, WI, USA) was used as solvent. Chemical shifts δ were reported in parts per million relative to the residual CHCl_3_ signal as an internal reference standard. Signal assignments are listed in [32,38]. The average lengths of NB and CDD sequences in the multiblock copolymer, *L*_NB_ and *L*_CDD_, were calculated from the integral intensities of homo- (NB–NB, CDD–CDD) and hetero- (NB–CDD, CDD–NB) dyad signals in the ^13^C NMR spectra as *L*_NB_ = [*I*(C=C_NB–CDD_) + *I*(C=C_NB–NB_)]/*I*(C=C_NB–CDD_) and *L*_CDD_ = [*I*(C=C_CDD–NB_) + *I*(C=C_CDD–CDD_)]/*I*(C=C_CO–NB_) (Figure 1). The growth of heterodyad content in the course of the reaction indicates a decrease in *L*_NB_ and *L*_CDD_, which are combined in the copolymer blockiness parameter, λ = 1 − 1/*L*_NB_ − 1/*L*_CDD_.

Standard DSC thermograms were recorded on a Mettler TA 4000 (Greifensee, Switzerland) system at a rate of 10 K/min under argon flow of 70 mL/min in the range from 193 to 373 K. A thermal successive self-nucleation and annealing (SSA) protocol common for two samples is shown in Figure 2. It is characterized by a higher temperature limit of 383 K set for 5 min, a standard crystallization temperature *T*_c_ = 233 K maintained for 2 min, and a set of self-nucleation temperatures *T*_si_ starting at 355 K down to 300 K with a step (fractionation window) of 5 K and a fractionation time of 19 min at each *T*_s_ and a cooling/heating rate of 20 K/min. Single-step annealing was performed by heating a sample up to 383 K and, 5 min later, cooling down to *T*_c_ set for 2 min, heating up to a temperature 5 K above the end of the melting curve for this sample, annealing for a given time (usually 30 min), cooling down to *T*_c_ set for 3 min, and then recording the heating thermogram, a cooling/heating rate of 10 K/min.

XRD patterns were recorded using an X-ray source with a rotating copper anode, Rotaflex RU-200 (Rigaku, Tokyo, Japan), with primary monochromatic CuKα radiation (λ = 1.542 Å), equipped with a horizontal goniometer, Rigaku D/Max-RC, in the scattering range (2θ) of 5°−55° at room temperature. A θ–2θ scanning of the sample’s films was carried out in the “reflection” geometry according to the Bragg–Brentano scheme. The measurement was carried out in continuous scanning mode at a speed of 1 degree/min and a step of 0.04 degrees. The diffractograms were processed using data from [52,53,54] for crystalline peak assignment. The lattice spacing, *d*, and crystallite size, *L*, were found for each peak from the Bragg law and Debye–Scherrer relation, respectively.

Rheological properties were measured in the bulk on a rotational rheometer, Physica MCR301 (Anton Paar, Graz, Austria), using a 25 mm-diameter plate–plate geometry with a 0.4 mm gap. Samples were loaded at 383 K and kept for 20 min to allow relaxation of the normal forces. Then, oscillatory experiments were carried out at a frequency of 1 Hz and a strain of 1% to probe the dependences of the storage modulus on temperature and time. In the first case, the samples were cooled down to 363 K, and the measurements were carried out in a temperature range above the melting temperature and below the onset of thermal oxidation at a heating/cooling rate of 5 K/min. In the second case, the samples were cooled down to a measurement temperature, then heated up to 383 K, annealed for 15 min, and cooled down to 293 K, inducing polymer crystallization. After 20 min, the samples were brought back to the measurement temperature.

## 3. Results and Discussion

### 3.1. Copolymer Synthesis and Chain Structure

Table 1 describes the cross-metathesis conditions and structural characteristics of PNB and PCDD homopolymers, their mixture, and seven NB–CDD C1–C7 multiblock copolymers synthesized from them. The copolymer molar mass is lower relative to the parent homopolymers, and it decreases by increasing the amount of the G1 catalyst in the initial polymer mixture. This can be explained by main-chain C=C bond cleavages due to their interaction with Ru-benzylidene carbenes of G1. Our kinetic study [38] demonstrates that this process takes place at the early stage of the cross-metathesis reaction, which lasts for about an hour.

Later on, the interaction between polymer-based carbenes and macromolecular double bonds results in the formation of a copolymer, which is gradually randomized with time without considerable changes in its molar mass. Table 1 demonstrates that the randomization rate depends on the initial PNB/PCDD/G1 mixture composition, in particular, on the component molar ratio, NB/CDD, and on the amount of catalyst, G1. The unit distribution in C1–C7 copolymers with the highest heterodyad content of 11% (C4 copolymer) is still far from the fully random one. The dyad fractions, *φ*_CDD-CDD_, *φ**_NB–CDD_ = *φ*_NB-CDD_ + *φ*_CDD-NB_, and *φ*_NB-NB_, are evaluated from the vinylene 128.0–136.0 ppm region of the ^13^C NMR spectra, an example of which is shown in Figure 1. They are used to calculate the fraction of CDD units, *φ*_CDD_ = *φ*_CDD-CDD_ + *φ*_CDD-NB_, and number average length of their blocks, *L*_CDD_ = *φ*_CDD_/*φ*_CDD-NB_, in NB–CDD copolymers, whereas for PCDD homopolymer, *L*_CDD_ is just equal to its degree of polymerization. Even for nonequimolar mixtures used for the synthesis of C5–C7 copolymers, the copolymer composition, *φ*_CDD_, is close to the composition of an initial mixture, NB/CDD molar ratio, rather than to the polymerization degree ratio of initial homopolymers. This indicates [12,55] that the homopolymer content in all copolymer C1–C7 samples is negligible.

### 3.2. Copolymer Structure and Crystallinity

Whereas PNB is amorphous, PCDD and NB–CDD copolymers are semicrystalline. Crystallinity in polyalkenamers is determined by *cis*/*trans* isomerism with respect to C=C main-chain double bonds [41]. As seen from Figure 1, the signals of (130.06 ppm) *cis* and (130.52 ppm) *trans* CDD–CDD homodyads are well resolved in the ^13^C spectra. Table 1 demonstrates that the highest *trans* content of 85% is found in PCDD synthesized with the G2 catalyst, whereas in NB–CDD copolymers obtained via the macromolecular cross-metathesis in the presence of G1, it slightly decreases to 77%–84%. As a result, wide-angle X-ray scattering patterns for the homopolymer and one of the copolymers (C1) shown in Figure 3 are qualitatively similar.

In both cases, the XRD peak analysis reveals the presence of monoclinic crystals characteristic of *trans* PCDD [52], with similar structural parameters listed in Table 2. The typical crystallite size is of the order of 100 nm, but it cannot be identified with the average lamella thickness, which is determined by the reflections from {00*l*} lattice planes that are absent in the present patterns. The degree of crystallinity, DC*_str_*, is lower in the NB–CDD copolymers than in the PCDD homopolymer. This can be explained by the fact that crystallizable *trans* CDD units, which are diluted with both *cis* CDD and amorphous NB units, form many nanosized crystallites poorly detectable with X-rays. Indeed, the number average length of *trans* CDD blocks in the NB–CDD copolymers, *L*_tr-CDD_, falls within the range 3.4 < *L*_tr-CDD_ < 5.4, which is less than the value of 6.7 estimated for PCDD from the NMR data (Table 1).

DSC provides an additional tool to monitor polymer crystallinity. Standard second-heat thermograms are shown in Figure 4 for all polymers except amorphous PNB. Table 3 contains characteristics of the studied polymers extracted from the DSC data. The degree of crystallinity according to DSC, DC_cal_, is found relative to the molar enthalpy of fusion for *trans* PCDD crystals, Δ*H*_µ_, which is equal to 38,460 and 37,620 J/mol at 85% and 80% *trans* C=C content, respectively [54].

As seen from Figure 4, the melting range spans across nearly 100 degrees. The maximums of C1–C7 copolymer melting curves are left-shifted to lower temperatures, reflecting a decrease in the average length of crystallizable *trans* CDD blocks, *L*_tr-CDD_, listed in Table 1. All the curves can be tentatively divided into two groups: (1) PCDD homopolymer, PCDD/PNB mixture, and C1, C2, and C5 copolymers with long (>30 units) CDD blocks and (2) C3, C4, C6, and C7 copolymers with relatively short (<20 units) CDD blocks.

Polymers from the first group possess high aspect ratio melting peaks with *T*_m_ above 345 K and DC_cal_ close to 50%. They become fully amorphous in a narrow 350–360 K range, which means that NB units do not markedly influence the formation of large PCDD crystals. The PCDD/PNB mixture reveals the highest *T_m_* of all samples, which indirectly indicates that it is phase separated. Note that DSC reports considerably higher degrees of crystallinity for PCDD and C1 copolymer than XRD (Table 2 and Table 3), which is less sensitive to small crystallites. Moreover, one can see from Figure 4 that melting starts well below room temperature at which X-ray scattering was performed.

Copolymers from the second group are characterized with left-shifted (*T*_m_ < 340 K) and less pronounced main endothermic peaks and weak secondary endotherms at even lower temperatures. Such behavior can be explained by decreasing block lengths and by constraints imposed by bulky and glassy NB blocks. A reader should not be misled by single *T*_g_ values for the NB–CDD copolymers in Table 3, which are obviously related to CDD blocks, whereas the glass transition for NB blocks is hidden somewhere in the 250–300 K range (*T*_g_ for PNB is 310 K) behind the melting peak.

Nevertheless, the degree of crystallinity per one *trans* CDD unit, DC_cal_, for C3 and C6 copolymers is close to 50%, as for the polymers of the first group. The remaining C4 and C7 copolymers reveal close *T_m_* values but a notably lower (ca. 35%) degree of crystallinity. Interestingly, C3 and C4 copolymers have a very similar chain structure (Table 1), and the apparent difference in their crystallinity should be attributed to the method of copolymer isolation. Whereas C3 copolymer, obtained in the presence of 1 mol.% of the G1 catalyst, was precipitated and then dried, C4 sample, which initially contained 5 mol.% of G1, was concentrated in vacuum and thus contained a considerable amount of the catalyst moiety. In the case of C7 copolymer, a drop in crystallinity can be related to the low (24%) content of CDD units in this sample, which probably do not form continuous domains. This version is consistent with a noticeable exothermic effect observed in the 275–305 K range for this copolymer, which typically reflects cold crystallization and indicates that the previous cooling was too fast for the crystallizable sequences to self-organize into large crystallites.

The crystal lamella thickness *L* can be roughly estimated using the Gibbs–Thomson equation [56]:(1)$$L=\frac{2\gamma }{\Delta H_{V}}\frac{T_{m0}}{T_{m0}-T}$$
where *γ* is the crystal surface free energy, Δ*H_V_* is the enthalpy of crystal fusion per unit volume, and *T_m_*_0_ is the equilibrium melting temperature of a homopolymer consisting of crystallizable copolymer units.

The value of *T_m_*_0_ = 357 K for *trans* PCDD crystals is given in [54], and Δ*H_V_* can be expressed in terms of the above-mentioned molar enthalpy of fusion, Δ*H*_µ_; density, ρ = 950 kg/m^3^; and molar mass of a repeating unit, µ = 0.166 kg/mol, as Δ*H_V_* = Δ*H*_µ_ρ/µ. Due to the lack of literature data on the surface free energy, we use a predictive approach linking *γ* to the chemical structure of a PCDD repeating unit. Two alternative group contribution expressions suggested by Askadskii [57] yield the values of 35.2 and 32.1 mJ/m^2^, which can be averaged into *γ* = 33.6 mJ/m^2^.

Substituting *T_m_* or *T_end_* for the temperature in Equation (1), we find the thickness of crystallites that melt near the maximum or end of the DSC curve, *L_m_* and *L_end_*, respectively (Table 3). Since the length of a CDD monomer unit in the crystal structure equals *a* = 14.85 Å [41], we obtain that for C3 and C4 copolymers, the ratio *L_m_*/*a* (3.57 and 3.37) is rather close to the number average length of the *trans* CDD sequence, *L_tr_*_-CDD_ (3.5 and 3.4, respectively). Indeed, in copolymers with rather short crystallizable blocks, chain folding is an unlikely event so that the average block length directly defines the typical size of the most common crystals. The largest crystallites that melt near *T_end_* vary in size from 8 nm for C4 copolymer to 210 nm for the PCDD/PNB blend. Such values are common for PCDD and polyalkenamers in general [41], for which *cis*/*trans* isomerism prevents the formation of micron-size crystallites typical of many polyolefins.

Macromolecular cross-metathesis is an interchange-type reaction that typically transforms a mixture of two homopolymers into a copolymer with the first-order Markov chain statistics [12]. Therefore, we can implement the Flory theory of statistical copolymer crystallization [58] to estimate the equilibrium melting temperature, $T_{m0}^{\left(c\right)}$, for all studied polymers, including PCDD, which can be considered a random copolymer of *trans* and *cis* CDD units:(2)$$T_{m0}^{\left(c\right)}={\left(\frac{1}{T_{m0}}-\frac{Rlnp}{\Delta H_{\mu }}\right)}^{-1}$$
where *p* is the conditional probability that a crystallizable unit is followed by a similar unit in a copolymer chain, and *R* is the molar gas constant. Assuming a random distribution of *cis* and *trans* C=C bonds, we can write *p* = *φ_tr_*_-CDD-*tr*-CDD_/*φ_tr_*_-CDD_ = *φ*_CDD-CDD_(*φ_tr_*_-CDD_/*φ*_CDD_)^2^, reducing the calculation of *p* to the information available from NMR data. As soon as *p* is known, $T_{m0}^{\left(c\right)}$ can be found using Equation (2). The values of *p* and $T_{m0}^{\left(c\right)}$ are listed in Table 3. It is seen that all predicted equilibrium melting temperatures belong to a rather narrow interval of 347–353 K due to relatively small variations in the parameter *p*, which are in turn conditioned by similar synthetic conditions. The difference between $T_{m0}^{\left(c\right)}$ and *T_m_* is highest for C4 and C7 copolymers, which means that their crystallization proceeds under less equilibrium conditions than in the case of copolymers with longer *trans* CDD blocks or PCDD homopolymer.

The experimental dependences of both maximum and end melting peak temperatures, *T_m_* and *T_end_*, on ln *p* can be fitted with straight lines (Figure 5) in the spirit of Equation (2) of the Flory theory. In the former case, the agreement is only qualitative, but in the latter situation, it is really good. This can be useful for predicting the values of *T_m_* and, especially, *T_end_* for NB–CDD copolymers with a known distribution of *trans* CDD groups. Nevertheless, the parameters of these lines are far from the reference values Δ*H*_µ_ = 38,460 J/mol and *T_m_*_0_ = 357 K for 85% *trans* PCDD [54] because standard DSC measurements are performed under nonequilibrium conditions.

### 3.3. Thermal Fractionation by DSC

More information on the crystallinity of a copolymer can be obtained by subjecting it to the successive self-nucleation and annealing procedure described in the Materials and Methods section and illustrated in Figure 2. Aiming to avoid thermochemical crosslinking at double C=C bonds, we started self-seeding at 355 K, which is 6 K above the end of the C1 copolymer melting curve. Then we decreased the annealing temperature to 300 K in 12 steps. Each step included cooling down to *T*_c_ = 233 K and backward heating to a temperature 5 K below the previous annealing. During such treatment, a set of crystallites with different melting temperatures is formed in the sample, thus leaving a footprint on its final heating thermogram. Figure 6 presents the melting curves for C1 and C4 copolymers, both with a number of endothermic peaks. One can see that the crystallites in C1 are characterized by a broader melting range than in C4. This distribution is dominated with high-melting crystallites in the former case and is much more uniform in the latter.

The attained separation of peaks in Figure 6 is enough for their deconvolution. By calculating areas under the peaks, one can characterize the distribution of polymer crystallites over their melting temperature (Figure 7a) or size (Figure 7b). The latter requires the application of the Gibbs–Thomson equation (Equation (1)). Since the differential distribution functions strongly fluctuate, we replaced them with integral ones that are plotted in Figure 7.

It follows from Figure 7b that, for both samples, the minimum crystallite thickness is 1.3 nm, which means that even isolated *trans* CDD units that are 1.485 nm long can be included in the crystallites. This situation is different from NB–COE copolymers [51], where a *trans* COE block has to contain at least two units to be incorporated into a crystallite. In other aspects, a pronounced difference between the copolymers with longer (C1) and shorter (C4) crystallizable blocks is evident. In the former case, about half of the crystalline domains are formed by large crystallites that are 8–15 nm thick, whereas the other copolymer does not contain crystallites of that size at all. Its crystallites are more uniformly distributed in thickness being bounded from above by ca. 8 nm. By comparing the distribution median positions in Figure 7b with the values of *L_m_* from Table 3, we find that applying the Gibbs–Thomson equation to a standard DSC trace considerably overestimates the average crystallite size. It is also worth noting that the toothed curve for C4 is similar to the corresponding curves describing NB–COE copolymers in [51], whereas the curve for C1 resembles that of PCOE homopolymer.

### 3.4. High-Temperature Endotherm

Second-heat DSC thermograms for NB–COE multiblock copolymers [51] revealed additional endothermic peaks that were markedly inferior to the melting peaks in area and shifted from them by 5 to 10 K toward higher temperatures. Similar weak endotherms with an area of several joules per gram were detected for completely amorphous NB–COAc (norbornene–5-acetoxy-1-cyclooctene) multiblock copolymers [34]. At the same time, the DSC curves for the corresponding homopolymers, semicrystalline PCOE and amorphous PCOAc, did not contain such endotherms. The observed heat absorption can be tentatively attributed to the mixing of norbornene- and octenylene-enriched melted domains separated at lower temperatures either due to their immiscibility or because of crystallization. It was interesting to check whether such high-temperature endotherms are detectable for the NB–CDD copolymers.

The routine calorimetry measurements (Figure 4) did not reveal anything worthy of attention, whereas the thermal fractionation of C1 copolymer possessing the longest blocks resulted in a weak endotherm on the final thermogram (Figure 6) near 370 K. In order to enhance the effect, we performed single-step annealing experiments, in which PCDD, its blend with PNB, and NB–CDD C1–C7 copolymers, after erasing the thermal history, were kept for 30 min at temperatures 5 K above the corresponding T_end_ values found from the standard DSC experiments and listed in Table 3. As seen from Figure 8a, the subsequent cooling–heating cycle reveals weak but noticeable endotherms for C1 and C6 copolymers located 8–10 K above the corresponding annealing temperatures.

Since C1 is the NB–CDD copolymer with the longest blocks, an endotherm appearance can be considered expectable, in agreement with the results in [51], where it was found that the endothermic effect diminishes by lowering the average length of alkenamer blocks. Figure 8b demonstrates that this peak becomes more pronounced and shifts to higher temperatures by increasing the annealing time. It does not appear without annealing of the sample above T_end_, subsequent deep cooling (in our study, down to the standard crystallization temperature T_c_ = 233 K), and reheating. It remains unclear why the high-temperature endotherm is found in C6 copolymer enriched with amorphous NB units and, at the same time, is absent in C5 copolymer of the nearly mirror composition with an excess of crystallizable CDD units.

The above thermal effect, although weak, about 1 J per gram (Figure 8), should affect the mechanical properties of NB–CDD copolymers in the melt state, which can be probed by rotational rheometry. Sensitive oscillatory experiments were performed with C1 and C7 copolymers after the single-step annealing, like in the calorimetric measurements. We began with the C7 sample, which did not reveal any high-temperature endotherms above its full amorphization temperature of 347 K. Figure 9a demonstrates that two subsequent heating–cooling cycles from 363 to 403 K and back, with annealing for 15 min at 363 K in between, lead to a smooth and reversible change in the storage modulus G′. However, an isothermal annealing of C7 copolymer at 349 K for an hour results in two different values of G′ depending on the sample history (Figure 9b). If it is cooled down to 349 K starting from the homogeneous melt at 383 K, then the modulus stays nearly constant (0.17 GPa) in the course of the annealing. However, if the copolymer is heated up to 349 K from the crystalline state at 293 K, then G′ slowly increases by 1.5 times to 0.24 GPa. It means that, immediately after melting, C7 copolymer undergoes a structural rearrangement between two inhomogeneous states, whereas a homogeneous morphology is stable at that temperature.

The situation with C1 copolymer, which possesses the high-temperature endotherm, is somewhat different. The first heating–cooling cycle does not return G’ to its original level, and only after 15 min of annealing at 363 K does the storage modulus take the value that can be recovered after the second heating and cooling (Figure 10a). In that case, an isothermal annealing at 359 K (6 degrees above T_end_) results in a slow growth of G’ irrespective of the previous temperature of the sample (Figure 10b). Such behavior indicates that above T_end_, neither melted-from-crystal nor homogeneous morphology of C1 copolymer is stable. The equilibrium could correspond to a microphase-separated state, which undergoes an order–disorder transition upon further heating. It is highly likely that this effect is detected as the high-temperature endotherm in the DSC experiments.

## 4. Conclusions

In this study, we investigated the thermal properties of novel multiblock copolymers of norbornene and cyclododecene and quantitatively related the structural parameters of polymer chains to the characteristics of crystallites formed by *trans* dodecenylene unit sequences. The copolymers with the number average length of such sequences exceeding 4.5 units crystallize similar to polydodecenamer and form crystalline lamellas up to 15 nm thick, whereas the copolymers with shorter *trans* dodecenylene blocks generate crystallites that are more uniform in size and not exceeding 8 nm. Thus, the information on the distribution of units in copolymer chains obtained by NMR can be used to predict their thermal properties. Vice versa, calorimetric data by DSC can be used to estimate the structure of macromolecules and to specify the synthetic conditions needed to attain the required degree of copolymer blockiness via the macromolecular cross-metathesis reaction. Such relations are crucial for the design of morphologies that arise via the self-assembly of multiblock copolymers.

Crystallization in copolymers with short dodecenylene blocks proceeds under less equilibrium conditions than that in copolymers with longer blocks or polydodecenamer homopolymers. Nevertheless, the application of thermal fractionation by successive self-nucleation and annealing allowed us to establish that even isolated *trans* dodecenylene units of ca. 1.5 nm length can be accommodated in polymer crystallites. In line with our previous studies, we found that the annealing of a multiblock copolymer above its full amorphization temperature can give rise to a weak but persistent endotherm 5–10 degrees above the annealing temperature. This effect is absent in the corresponding homopolymers and may present an experimental evidence of microphase separation in multiblock copolymers of different chemical nature.

## Data Availability

The data presented in this study are available on request from the corresponding author.

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
