# Peer review of "Multiblock Copolymers of Norbornene and Cyclododecene: Chain Structure and Properties"

_polymers, 2021, doi:10.3390/polym13111756_

Round 1

Reviewer 1 Report

this is an interesting manuscript and the authors did a good job present the results. Thus it is recommended to publish in Polymers after after addressing the following minor points:

(1) molecular weight plays key roles in the properties of these polymers. However, the data shown here were all apparent MW from GPC with RI detector. Results from LS data should be included.

(2) Some English descriptions could be improved. Please have a native speaker proofread it.

Author Response

This is an interesting manuscript and the authors did a good job present the results. Thus it is recommended to publish in Polymers after after addressing the following minor points:

(1) molecular weight plays key roles in the properties of these polymers. However, the data shown here were all apparent MW from GPC with RI detector. Results from LS data should be included.

Response: It is hard to agree with this remark because the most important parameters, which determine various properties of high-molar-mass copolymers, are their average block lengths determined from the NMR data. We in fact performed GPC experiments with only one (RI) detector since we wanted to just monitor relative changes in the MM in the course of the macromolecular cross-metathesis reaction. Moreover, when we have synthesized such copolymers for the first time, we indeed tried to use GPC with a LS detector but encountered technical problems related to different solvent affinity toward polymer blocks (and such problems are typical, as follows from the recent review in Polym. Chem. 12 (2021) 2522)

 (2) Some English descriptions could be improved. Please have a native speaker proofread it.

Response: We followed the advice of this referee to correct several misprints and stylistic inconsistencies.

Reviewer 2 Report

The authors investigate structure-property relations for the multiblock copolymers of norbornene with cyclododecene synthesized via the macromolecular cross-metathesis reaction between amorphous polynorbornene and semicrystalline polydodecenamer in the presence of the 1st generation Grubbs catalyst. Structural, thermal and mechanical characterization of the copolymers were performed by NMR, XRD, DSC techniques. Some interest results have been obtain by this study. And the paper has been well-edited. I suggest that this paper can be accepted for publication after a minor revision.

(1) When abbreviations are first used, please provide their full names.

(2) Some figures shall be revised for good presentation.

(3) What is the mean of “00l” in the Line 280?

(4) Ltr-CDD or LCDD-tr, please select one of them.

(5) The references shall not be used in the “Conclusions”.

(6) “Conclusions” needs to be rewritten for revealing the main results of this work.

Author Response

The authors investigate structure-property relations for the multiblock copolymers of norbornene with cyclododecene synthesized via the macromolecular cross-metathesis reaction between amorphous polynorbornene and semicrystalline polydodecenamer in the presence of the 1st generation Grubbs catalyst. Structural, thermal and mechanical characterization of the copolymers were performed by NMR, XRD, DSC techniques. Some interest results have been obtain by this study. And the paper has been well-edited. I suggest that this paper can be accepted for publication after a minor revision.

(1) When abbreviations are first used, please provide their full names.

Response: This was corrected for DSC, XRD, GPS, and SSA abbreviations.

(2) Some figures shall be revised for good presentation.

Response: Figures 1, 2 and 8 were improved and replaced with new versions.

(3) What is the mean of “00l” in the Line 280?

Response: 00l means a set of lattice planes which can be useful to estimate the average thickness of crystalline lamellas from XRD data. The corresponding sentence in the last paragraph of page 6 was extended.

(4) Ltr-CDD or LCDD-tr, please select one of them.

Response: We introduced corrections in favor of Ltr-CDD

(5) The references shall not be used in the “Conclusions”.

Response: Two references were removed from the Conclusions section

(6) “Conclusions” needs to be rewritten for revealing the main results of this work.

Response: We believe that Conclusions already contained main results of the study but introduced more details in the first paragraph of this section.

Reviewer 3 Report

In my opinion the submitted work entitled “„Multiblock copolymers of norbornene and 2 cyclododecene: Chain structure and properties” may be published.

This is, in general, an interesting and valuable work.

The Authors investigated the thermal properties of novel multiblock copolymers of norbornene and cyclododecene and quantitatively related the structural parameters of polymer chains to the characteristics of crystallites formed by trans dodecenylene unit sequences.

The results are well documented, and the conclusions are of interest of many researches and practicioners.

I did not notice any editorial errors in the work, I have no comments about the form of the results presented. Only one small notification: due to the lack of reference to Scheme 1 in the text, the Authors should complete it.

Author Response

No critical comments from this reviewer. Parts a and b of Scheme 1 are mentioned in the paper text in two paragraphs of the Materials and Methods section preceding this scheme (page 3).